# Hierarchical Molecular Graph Self-Supervised Learning for property prediction

Xuan Zang [1], Xianbing Zhao[1] & Buzhou Tang[1,2]✉

Molecular graph representation learning has shown considerable strength in molecular analysis and drug discovery. Due to the difficulty of obtaining molecular property labels, pre-training models based on self-supervised learning has become increasingly popular in molecular representation learning. Notably, Graph Neural Networks (GNN) are employed as the backbones to encode implicit representations of molecules in most existing works. However, vanilla GNN encoders ignore chemical structural information and functions implied in molecular motifs, and obtaining the graph-level representation via the READOUT function hinders the interaction of graph and node representations. In this paper, we propose Hierarchical Molecular Graph Self-supervised Learning (HiMol), which introduces a pre-training framework to learn molecule representation for property prediction. First, we present a Hierarchical Molecular Graph Neural Network (HMGNN), which encodes motif structure and extracts node-motif-graph hierarchical molecular representations. Then, we introduce Multi-level Self-supervised Pre-training (MSP), in which corresponding multi-level generative and predictive tasks are designed as self-supervised signals of HiMol model. Finally, superior molecular property prediction results on both classification and regression tasks demonstrate the effectiveness of HiMol. Moreover, the visualization performance in the downstream dataset shows that the molecule representations learned by HiMol can capture chemical semantic information and properties.

[1] Department of Computer Science, Harbin Institute of Technology, 518055 Shenzhen, China. [2] Pengcheng Laboratory, 518055 Shenzhen, China.
✉email: tangbuzhou@gmail.com

n recent years, molecular representation learning has attracted growing attention in the study of chemical drug analysis and discovery[1]. Motivated by the remarkable success of machine learning[2], especially deep learning[3], molecular representation learning has increasingly explored machine learning-based methods. These methods are widely used for various molecular applications, such as chemical property prediction[4,5], drug molecule generation[6–8], and optimization[9]. How to learn comprehensive and effective molecular representation remains an open challenge.

Inspired by the development of Natural Language Processing (NLP)[10–12], a range of methods apply language models to handle string-based molecular representations like SMILES[13] and SELFIES[14]. Specifically, these molecular strings are encoded by Recurrent Neural Network (like Gated Recurrent Unit (GRU)[15] and Long Short-Term Memory (LSTM)[16] or Transformer[12], which can be trained in a supervised manner by predicting molecular properties[17]. Further, to handle enormous unlabeled data, many works pay attention to self-supervised learning (SSL) frameworks. The SSL decoders generally generate the original strings[18] or other random SMILES of input molecules[19] or recover masked tokens of molecular strings[20]. Due to the single dimension of strings, important topology structures of molecular graphs are ignored. Therefore, many researchers have transformed their interests from 1D molecular strings to 2D graphs[21,22]. In view of the success of SSL, graph SSL-based pre-training frameworks[23–27] have been increasing rapidly in the past few years. They capture the topology of 2D graphs and have also shown a positive effect in molecular analysis tasks. However, their commonality for all kinds of graphs leads to the neglect of unique structural properties of chemical molecules, such as rings and functional groups.

Many recent presentation learning works[5,22,28–33] consider the characteristic of molecular graphs. Zhang et al.[32] model a clustering problem to learn molecular motifs, and the GNN-encoded atom representations are grouped into subgraphs for contrastive learning. Zhang et al.[30] design self-supervised molecular pre-training framework, whose pretext task is predicting the motifs based on a given order (depth-first search or breadth-first search) on the graph. Wang et al.[5] present three molecular graph structural augmentation patterns and aligns different augmentations of the same molecule through contrastive learning. Wang et al.[33] also propose a contrastive learning framework for molecular learning, in which not only molecular pairs but also motif pairs are sampled for contrastive pre-training. Wang et al.[31] takes into account the chemical reaction relationships between the molecules, i.e., the sum of molecular representations of all reactants is supposed to equal that of all products. Despite the improvement of molecule representation learning, previous works still fail to solve the following

problems and challenges: (1) How to preserve and capture molecular structure adequately? Many recent molecular learning methods[5,28] apply graph augment to construct different views and contrast multiple views for pre-training. However, some general graph data augmentations (like edge modification[34] or graph diffusion[35]) tend to destroy the structure or attributes of molecules, so some important chemical properties are highly possible to be buried. In addition to the whole topology structure, the motif is also valuable and has an important impact on the molecular properties. Therefore, it is beneficial and challenging for molecular learning to preserve the complete molecular structure and directly incorporate the motifs. (2) How to fuse more comprehensive information into molecular graph representations? With the development of GNN[36–40], growing molecular learning methods[29,31] leverage GNN as the encoder backbone, in which nodes flow information among local neighbors, and the graph representation is obtained by integrating all its node representations via READOUT function. The pattern of aggregating neighborhood representations lacks a global scale. Moreover, the READOUT operation can only transfer information from low-order atoms to the high-order molecule, but can not implement the interaction between them. (3) How to design the pretext tasks of self-supervised pre-training? Self-supervised pre-training requires design pretext tasks to optimize the backbone model parameters. The pretext tasks affect the transferable ability of the pre-training model to a great extent, which determines whether the model can achieve satisfactory fine-tuning performance in downstream tasks. Therefore, it is key for self-supervised pre-training models to design reasonable pretext tasks to improve transferable ability.

To resolve the aforementioned challenges, we propose a self-supervised learning framework for molecular property prediction called Hierarchical Molecular Graph Self-supervised Learning (HiMol). Figure 1 shows the overall framework of HiMol. HiMol consists of two major components, Hierarchical Molecular Graph Neural network (HMGNN) and Multi-level Self-supervised Pre-training (MSP). HMGNN presents a GNN-based hierarchical molecule encoder, aiming at addressing the first two challenges. To better explore the unique internal chemical structure of the molecular graphs, we improve rules to decompose the molecular motif structure. The molecular motif denotes the substructure of a molecule, which contains many chemical characteristics in general. For instance, the carboxyl group usually indicates acidity. The motif decomposition rules conform to chemical criteria to avoid destroying the chemical characteristics. We add motifs into the molecular graph as nodes to mine implicit semantic information. In addition, we augment a graph-level node to learn molecular graph representation through training along with normal nodes and motifs,

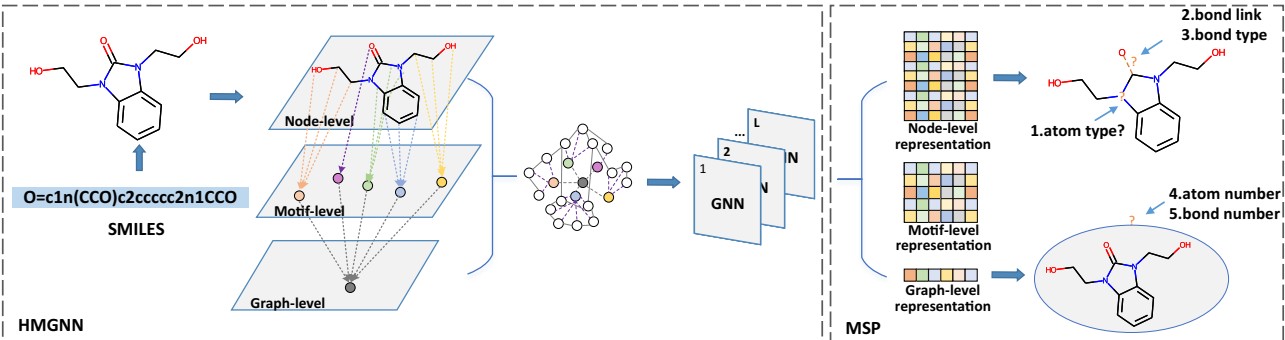

**Fig. 1 Illustration of HiMol.** HMGNN: the input molecular graph is first decomposed into motifs, which are constructed as motif-level nodes. Further, a graph-level node is added. The augmented graph is constructed by adding node-motif and motif-graph edges. GNN is employed to learn hierarchical node representations on the augmented graph. MSP: encoded atom representations and molecular graph representation predict five tasks for self-supervised pre-training.

in place of the READOUT function. The graph-level node collects information from all nodes and transfers the global information back to nodes through motif-level nodes. This circulation realizes the information interaction between different orders, which helps the model to learn more comprehensive representations. MSP designs multi-level pretext tasks as supervision signals of pre-training. Specifically, MSP designs three generative tasks that predict bond links, atom types, and bond types with the atom representations; MSP designs two predictive tasks that predict the number of atoms and bonds with the molecule representation. Compared with single-signal pre-training frameworks, multi-level pretext tasks help fuse more molecular information into molecule representations from diverse perspectives.

In summary, the advantages of our HiMol are as follows:

- HiMol builds a hierarchical GNN to encode molecules. It facilitates incorporating multi-order information into the molecule representation. To the best of our knowledge, HiMol is the first molecular representation learning method that encodes node-motif-graph hierarchical information.
- HiMol constructs motifs based on chemical rules and mine the characteristics of molecular substructures without destroying chemical structures.
- HiMol augments a graph-level node to simulate the molecular graph representation. It allows the molecular graph representation to participate in the training directly and realizes bidirectional transmission of local and global features.
- HiMol designs multi-level self-supervised pre-training tasks in accordance with hierarchical representations of molecules. It improves the transferable potential of the pre-training framework and helps learn more informative molecular representations.
- HiMol outperforms the state-of-the-arts (SOTA) in the downstream molecular property prediction tasks including classification and regression. It demonstrates the effectiveness of our proposed HiMol.

## Results

**HiMol framework**. Our HiMol presents a self-supervised pre-training framework to encode hierarchical representations of the molecular graph. HiMol mainly includes the Hierarchical Molecular Graph Neural network (HMGNN) and Multi-level Self-supervised Pre-training (MSP).

Given the SMILES of a molecule, we first transform it into a graph $G = (V, E)$ based on RDKIT[41], where $V$ and $E$ denote atom and bond sets respectively. Then we decompose $G$ into a series of motifs $V_m = \{V_m^1, V_m^2, ..., V_m^k\}$, $k$ is the number of motifs. The decomposition of motifs adds a rule on the basis of BRICS[42], i.e., break large ring fragments and select their minimum rings as generated motifs. All motif nodes $V_m$ are incorporated into $V$, and edges between the corresponding atoms and motifs $E_m$ are linked and merged into $E$. In addition, we augment a graph-level node $V_g$, meanwhile, edges between all motifs and $V_g$ are appended into the initial graph $G$. The augmented graph $\widetilde{G}$ is input into GNNs to learn the hierarchical presentations, including atom-level representations $\mathbf{H}_a \in \mathbb{R}^{|V| \times d}$, motif-level representations $\mathbf{H}_m \in \mathbb{R}^{k \times d}$, and graph-level representation $\mathbf{H}_g \in \mathbb{R}^d$. In the process of MSP, the atom-level representations are used to predict atom types, bond links, and bond types; the graph-level representation is used to predict the number of atoms and bonds. Cross entropy loss and smooth L1 loss are respectively applied to optimize the atom- and molecule-level learning.

As for the fine-tuning for downstream tasks, the graph-level representations pass through a 2-layer Multi-Layer Perceptron (MLP) to predict molecular properties. The pre-trained GNN weights are transferred to the fine-tuning model and continue to be updated along with the MLP parameters under the supervision of labels. More details about HiMol method are shown in "Methods".

**Molecular property prediction**. To evaluate the effectiveness of HiMol for molecular property prediction, we conduct classification and regression tasks on diverse datasets from MoleculeNet. Table 1 reports the mean and standard deviation of ROC-AUC (%) results on the binary classification tasks. We summarize the observations from the results as follows. (1) Our HiMol achieves the strongest performance on four out of six datasets. For the remaining datasets, HiMol still reaches competitive performance on Tox21. Although MolCLR shows remarkable results on ClinTox, its performance on other datasets is not satisfactory. On average, HiMol improves by 2.4% over the best-performing baseline. The enhancement proves the superiority and effectiveness of our hierarchical graph learning model HiMol. (2) Among

**Table 1 Molecular property prediction performance on classification benchmarks.**

| Datasets | BACE | BBBP | Tox21 | ToxCast | SIDER | ClinTox | Avg. |
|---|---|---|---|---|---|---|---|
| GraphSAGE | 72.7 ± 3.3 | 67.7 ± 2.8 | 69.9 ± 1.1 | 59.1 ± 0.3 | 58.3 ± 0.2 | 52.1 ± 5.5 | 63.3 |
| GPT_GNN | 72.5 ± 0.8 | 69.3 ± 1.3 | 73.1 ± 0.7 | 59.8 ± 0.4 | 59.6 ± 3.5 | 60.4 ± 3.3 | 65.8 |
| AttributeMask | 80.1 ± 0.4 | 65.9 ± 1.3 | 74.6 ± 0.3 | 63.7 ± 0.4 | 58.2 ± 0.6 | 74.0 ± 2.3 | 69.4 |
| ContextPred | 77.7 ± 1.3 | 68.6 ± 0.9 | 72.7 ± 0.6 | 62.1 ± 0.4 | 58.8 ± 1.1 | 71.1 ± 3.6 | 68.5 |
| InfoGraph | 76.6 ± 1.8 | 68.8 ± 0.7 | 74.7 ± 0.4 | 60.8 ± 0.8 | 56.7 ± 0.9 | 72.9 ± 4.7 | 68.4 |
| MoCL | 75.1 ± 0.1 | 66.8 ± 0.1 | 70.9 ± 0.2 | 60.7 ± 0.1 | 61.2 ± 0.1 | 60.8 ± 0.1 | 65.9 |
| GraphLoG | 79.0 ± 0.7 | 65.7 ± 1.4 | 73.4 ± 0.3 | 63.4 ± 0.4 | 57.3 ± 2.3 | 72.5 ± 1.8 | 68.6 |
| GraphCL | 72.8 ± 5.4 | 69.5 ± 2.6 | 75.0 ± 0.3 | 63.2 ± 0.4 | 61.4 ± 1.3 | 78.9 ± 4.2 | 70.1 |
| JOAO | 72.2 ± 2.0 | 70.7 ± 0.6 | 75.5 ± 0.7 | 61.6 ± 0.6 | 61.1 ± 0.9 | 79.6 ± 3.7 | 70.1 |
| MolCLR | 76.5 ± 0.5 | 69.3 ± 0.5 | 74.2 ± 0.6 | 55.0 ± 1.3 | 56.4 ± 1.7 | **90.4 ± 1.7** | 70.3 |
| G_Motif | 81.1 ± 3.2 | 68.6 ± 2.5 | 73.3 ± 0.8 | 61.0 ± 0.7 | 59.8 ± 1.3 | 78.9 ± 1.4 | 70.5 |
| MGSSL | 79.1 ± 0.9 | 69.7 ± 0.9 | **76.5 ± 0.3** | 64.1 ± 0.7 | 61.8 ± 0.8 | 80.7 ± 2.1 | 72.0 |
| HiMol (SMALL)[a] | **84.6 ± 0.2** | 71.3 ± 0.6 | 76.0 ± 0.2 | 66.0 ± 0.2 | **62.5 ± 0.3** | 70.6 ± 2.1 | 71.8 |
| HiMol (LARGE)[b] | 84.3 ± 0.3 | **73.2 ± 0.8** | 76.2 ± 0.3 | **66.3 ± 0.4** | 61.3 ± 0.5 | 80.8 ± 1.4 | **73.7** |

[a]SMALL version implements 3-layer GIN as the GNN backbone.
[b] LARGE version implements 5-layer GIN as the GNN backbone.
The values in bold highlight the best performing results of each benchmark.
Means and standard deviations of test ROC-AUC (%) are reported.

all SOTA methods, two motif-based models G_Motif and MGSSL outperform other baselines, which indicates the capture of motif structure play an important role in molecular graph learning. However, they only use motif prediction as the pretext task of pre-training, the hidden representations extracted by GNNs do not directly incorporate motif information during the fine-tuning for the downstream tasks. Our HiMol encodes motif structures through the GNN backbone, which benefits the molecular representation to better learn and predict the internal properties of molecules during fine-tuning. (3) We give two different scales of HiMol models, in which the layer numbers of GIN for HiMol (SMALL) and HiMol (LARGE) are respectively 3 and 5. The LARGE version performs slightly better, which may be explained as the deeper GNN being able to capture more structural information.

The regression results are shown in Table 2. According to the recommendation of MoleculeNet, mean-absolute error (MAE) is used as an evaluation metric for physical chemistry datasets (ESOL, FreeSlov, and Lipophilicity) while root-mean-square error (RMSE) is the metric for quantum mechanics datasets (QM7, QM8, and QM9). The observations of Table 2 are summarized as follows. The regression result of MoCL[28] on QM9 is not given since it is expensive to calculate the similarity matrix in the process of pre-training on QM9, and no corresponding result is given in the official paper. (1) HiMol outperforms all baselines on five out of six datasets and reaches competitive performance on the remaining QM8. Notably, the MAE decreases by 55.5% over the strongest baseline on the challenging dataset QM9. (2) Similar to the results of classification tasks, the LARGE version has a small edge over the SMALL version in the regression tasks.

**Visualization of molecular representations**. To exhibit intuitively the learned representations by HiMol, we utilize t-SNE[43] to project them to a two-dimensional space and different colors to distinguish molecular property labels. Note that we simply visualize the fine-tuned molecular presentations in the test set, which do not incorporate groundtruth. From Fig. 2, molecular presentations are grouped in line with their labels on both classification and regression tasks. For example, in Fig. 2a, the Class label represents the binary results of binding for BACE-1 inhibitors. Molecules labeled 0 and 1 are grouped on the top left and bottom right, respectively. Similarly, in Fig. 2b, molecules have increasingly high exp values from the top left to bottom right.

Moreover, in Fig. 2c, d, it can be observed that the visualized presentations have a high degree of consistency in terms of two important properties of molecules on QM9. The visualization in Fig. 2 demonstrates that our HiMol can extract the internal properties of molecules. Similar conclusions can be observed in other datasets (see Supplementary Fig. 1).

To further demonstrate the effectiveness of our pre-training framework, we visualize the molecular representations fine-tuned in downstream tasks without pre-training in Supplementary Fig. 2. Compared with Fig. 2, the visualization results without pre-training are chaotic in terms of molecules with different properties. The pre-training improves the performance of the HMGNN encoder and is beneficial to learning more chemical properties in downstream tasks.

**Molecular representation similarity ranking**. To further investigate the semantic information contained in the pre-trained molecular representations via HiMol, we evaluate the similarity between the query molecule and other molecules and draw the top-five most similar molecules. The cosine similarity between the query molecule $q$ and the candidate molecule $c$ is calculated as follows:

$$\text{sim}(q, c) = \frac{\mathbf{H}_g^q \cdot \mathbf{H}_g^c}{\left\|\mathbf{H}_g^q\right\|\left\|\mathbf{H}_g^c\right\|} \tag{1}$$

where $\mathbf{H}_g^q \in \mathbb{R}^d$ and $\mathbf{H}_g^c \in \mathbb{R}^d$ are the graph representations of the query molecule and candidate molecule, respectively. Then the similarity of all candidate molecules is ranked and the top-five molecules of the query ZINC9452931 are displayed in Fig. 3. From the figure, we can observe that the five molecules have similar structures and functional groups with the query. Specifically, the top-five similar molecules possess all atom types of the query, i.e., C, H, N, O, S. Moreover, the structure that two rings share a common edge exists in all molecules. Top@1 molecule not only have similar ring structures with the query, but also the same chain structures like NC(=O) and C(=O). The molecular representation similarity ranking experiments imply that our framework HiMol can learn molecular representation with chemical semantic information. Meanwhile, encoding the motifs is conducive to identifying molecular substructures like functional groups and rings. More results about other query molecules are shown in Supplementary Fig. 3.

**Table 2 Molecular property prediction performance on regression benchmarks.**

| Datasets | ESOL | FreeSolv | Lipophilicity | QM7 | QM8 | QM9 |
|---|---|---|---|---|---|---|
| Metrics | RMSE | RMSE | RMSE | MAE | MAE | MAE |
| GraphSAGE | 2.575 | 5.051 | 1.212 | 164.062 | 0.0388 | 11.178 |
| GPT_GNN | 1.612 | 5.284 | 0.820 | 229.053 | 0.0204 | 7.976 |
| AttributeMask | 1.439 | 8.062 | 0.784 | 261.588 | 0.0188 | 13.461 |
| ContextPred | 1.430 | 8.616 | 0.838 | 243.551 | 0.0205 | 16.886 |
| InfoGraph | 1.380 | 31.118 | 0.926 | 292.601 | 0.0192 | 12.350 |
| MoCL | 1.425 | 3.233 | 0.998 | 198.215 | 0.0903 | NA |
| GraphLoG | 1.390 | 4.515 | 0.857 | 274.071 | 0.0193 | 11.484 |
| GraphCL | 1.265 | 5.569 | 0.782 | 285.967 | 0.0199 | 9.773 |
| JOAO | 1.355 | 4.280 | 0.771 | 270.839 | 0.0206 | 22.507 |
| MolCLR | 1.333 | 3.285 | 0.720 | 104.184 | **0.0187** | 23.226 |
| G_Motif | 1.286 | 4.432 | 0.779 | 222.957 | 0.0203 | 11.065 |
| MGSSL | 1.346 | 2.980 | 0.751 | 155.913 | 0.0198 | 21.538 |
| HiMol(SMALL)[a] | 0.938 | 3.215 | 0.709 | 96.776 | 0.0196 | 3.770 |
| HiMol(LARGE)[b] | **0.833** | **2.283** | **0.708** | **91.501** | 0.0199 | **3.243** |

[a]SMALL version implements 3-layer GIN as the GNN backbone.
[b]LARGE version implements 5-layer GIN as the GNN backbone.
The values in bold highlight the best performing results of each benchmark.

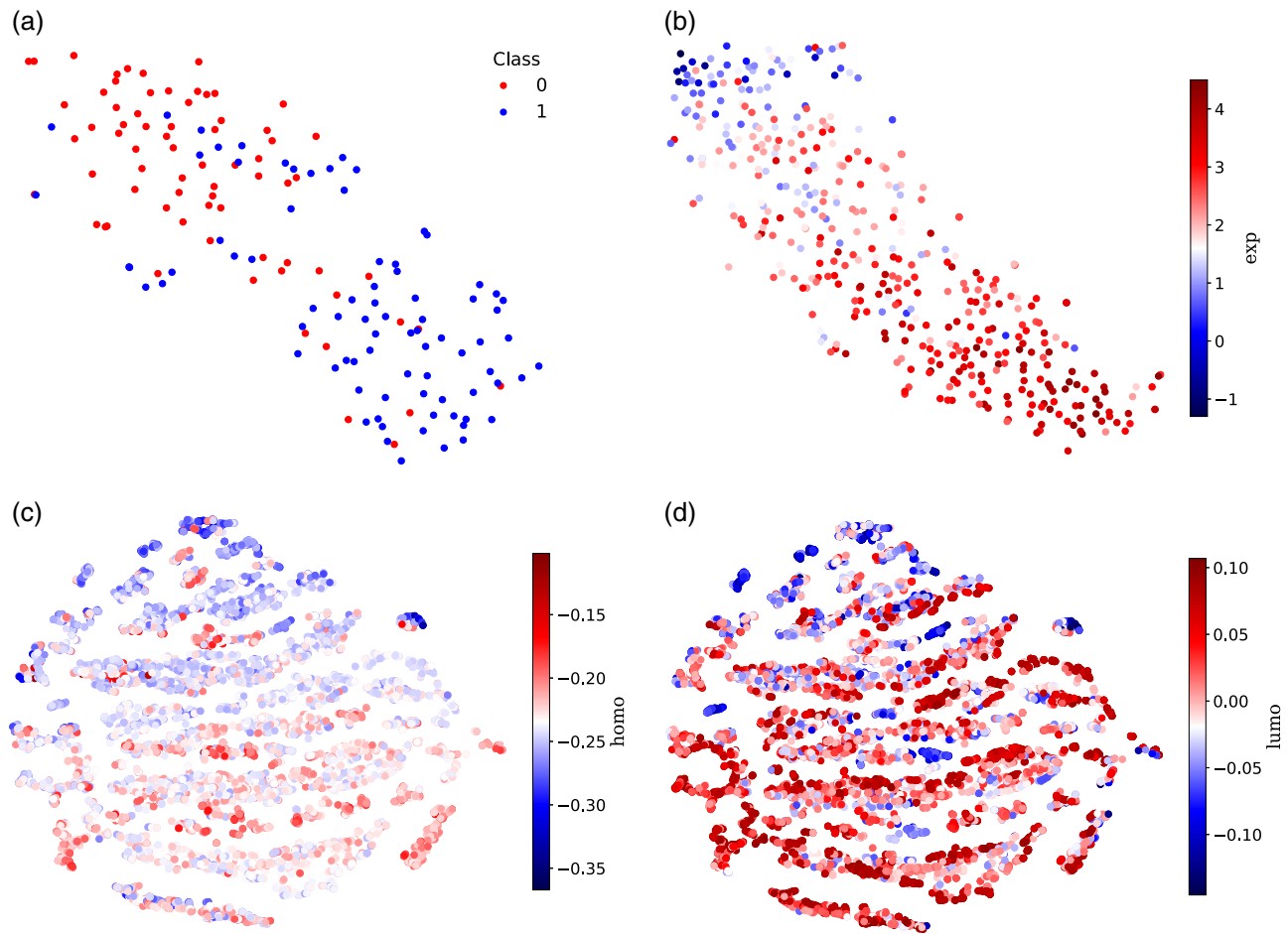

**Fig. 2 Visualization of molecular representations obtained by our HiMol on the downstream test set. a** BACE (Class): color represents binary labels of binding results for BACE-1 inhibitors. **b** Lipophilicity (exp): color represents octanol/water distribution coefficient. **c** QM9 (homo): color represents highest occupied molecular orbital energy (homo) of molecules. **d** QM9 (lumo): color represents lowest unoccupied molecular orbital energy (lumo) of molecules.

**Fine-tuning mode analysis**. To investigate the impact of different fine-tuning strategies on the model prediction tasks, we explore freezing mode and full fine-tuning mode. They both inherit the pre-trained weights of the hierarchical molecular graph neural network. The freezing mode freezes the parameters of HMGNN and only updates the weights of downstream classifiers, while the full fine-tuning mode trains all weights of HMGNN and classifiers in an end-to-end manner during fine-tuning. In addition, we evaluate HiMol without pre-training and vanilla GIN[38], which are directly trained through supervised downstream tasks. Table 3 shows two different versions of all methods, where SMALL version and LARGE version encode molecular graphs by 3-layer and 5-layer GNNs, respectively. From the results, we can observe that: (1) Full fine-tuning mode presents more advantage over freeze mode, which indicates that our proposed HMGNN plays a decisive role for prediction tasks; (2) Himol without pre-training exhibits better performance to GIN, which demonstrates that HMGNN has the ability to extract more abundant structural information; (3) HiMol achieves better performance after pre-training, which verifies that our proposed multi-level self-supervised pre-training strategy can effectively train model parameters and transfer knowledge to downstream data.

Figure 4 further displays the train and test classification ROC-AUC curves of the four aforementioned modes. It can be observed that the full HiMol can converge faster to optimum performance than other methods. Besides, the pre-trained methods (HiMol and its freeze mode) have better stability than the other two modes without

pre-training in terms of training and testing curves, indicating that our pre-training framework boosts the stability of fine-tuning.

**Ablation studies on pre-training**. To analyze the effectiveness of each part of our proposed HiMol framework, we conduct ablation experiments on HMGNN and MSP during the pre-training process. Figure 5 illustrates the ablation experimental results of HMGNN and MSP.

In Fig. 5a, we compare HiMol with two pre-training versions. HMGNN w/o motif-level denotes that molecular graphs only augment graph-level nodes and node-graph edges, without decomposing motifs. HMGNN w/o graph-level obtains graph-level representations through performing mean pooling to motif-level presentations. We can observe the following insights: (1) The performance of HiMol removing motifs becomes worse on all benchmarks, which implies the important role of encoding motifs for molecular representation learning. HiMol decomposes motifs to learn hierarchical representations, which is beneficial to learn more molecular structural characteristics and functions. (2) Replacing the graph-level node representation with the squeezed representation via graph pooling also makes the molecular property prediction performance slightly decrease. Augmenting a graph-level node is superior to the mean-pooling function, since it can indirectly learn multi-order features through the motif node, and can transfer the global information to all atom nodes during training. Mutually, atom nodes not only integrate local neighbor information but also capture substructural and global graph information. These multi-level

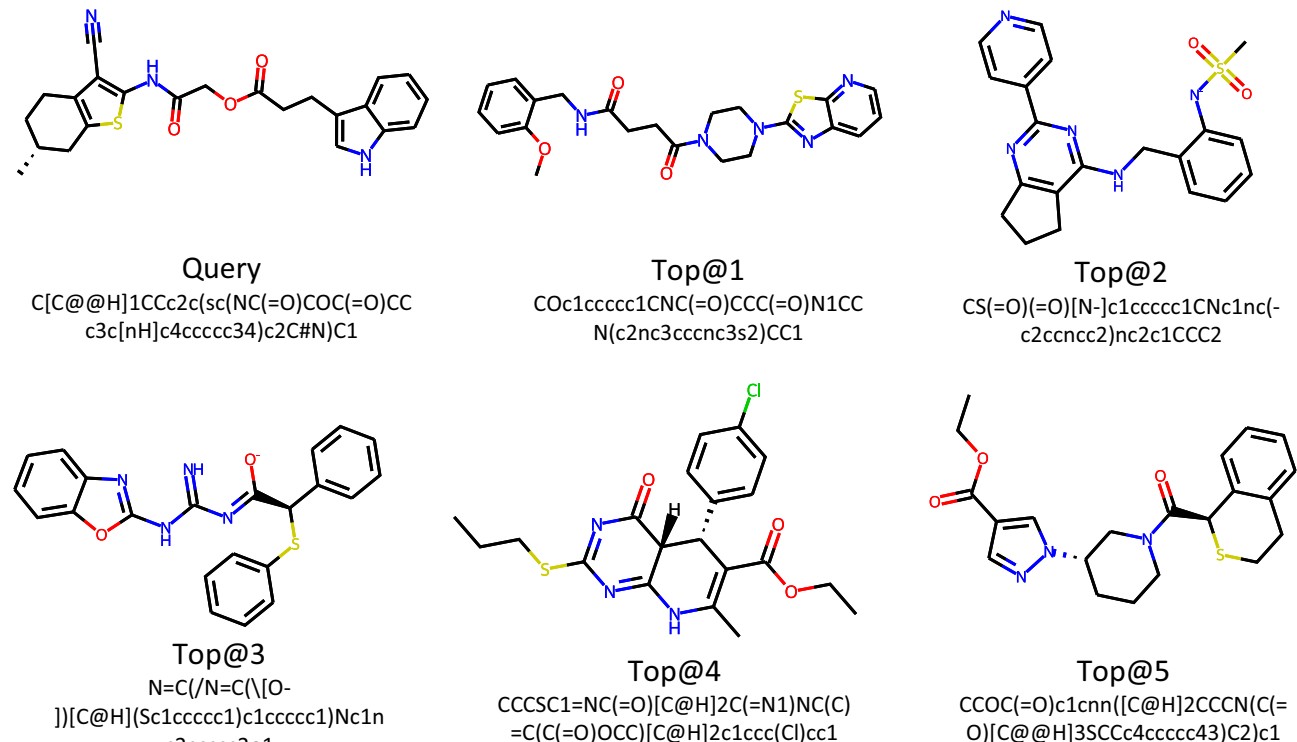

**Fig. 3 Visualization of the top-five molecules ranked by molecular representation similarity for the query ZINC9452931.** SMILES for all molecules are given.

**Table 3 Classification performance of different fine-tuning modes.**

| Methods | Version | BACE | BBBP | Tox21 | ToxCast | SIDER | ClinTox |
|---|---|---|---|---|---|---|---|
| GIN | SMALL | 72.6 | 70.2 | 72.2 | 62.7 | 57.5 | 60.7 |
| | LARGE | 73.4 | 67.7 | 73.8 | 62.5 | 56.2 | 60.8 |
| HiMol (no pre-train) | SMALL | 83.3 | 67.5 | 74.1 | 64.4 | 58.7 | 60.0 |
| | LARGE | 79.6 | 68.4 | 74.7 | 63.2 | 62.1 | 62.6 |
| HiMol (freeze) | SMALL | 77.9 | 59.2 | 68.1 | 60.9 | 61.1 | 66.0 |
| | LARGE | 72.6 | 60.3 | 65.4 | 59.3 | 61.5 | 67.6 |
| HiMol | SMALL[a] | **84.6** | 71.3 | 76.0 | 66.0 | **62.5** | 70.6 |
| | LARGE[b] | 84.3 | **73.2** | **76.2** | **66.3** | 61.3 | **80.8** |

[a]SMALL version implements 3-layer GIN as the GNN backbone.
[b]LARGE version implements 5-layer GIN as the GNN backbone.
The values in bold highlight the best performing results of each benchmark.

interactions between normal nodes and the graph-level node improve the learning performance of HMGNN.

The ablation versions for MSP are shown in Fig. 5b. MSP w/o atom-level contains only graph-level pretext tasks in the pre-training (the number of atoms and bonds). Similarly, MSP w/o graph-level contains only atom-level pretext tasks (bond links, atom types, and bond types). MSP w/o $\alpha$ simply sums all the loss values, removing the $\alpha$ weight parameter to balance. It can be observed that the combination of multi-level self-supervised tasks promotes the comprehensiveness transfer ability of pre-training compared with a single pretext task. In addition, the learnable weight $\alpha$ adjusts the importance of the loss of different parts, which can better coordinate the relationship between multi-level tasks to improve the performance of pre-training.

### Discussion
**Motif encoding**. First of all, we propose to encode the constructed motifs while learning the atom representations. HiMol

mines the substructure of molecules through motifs. There are many existing molecular representation learning methods[29,30] considering motif structure. They generally decompose molecules in a large-scale database into motifs and build a motif dictionary, then motif prediction serves as self-supervised pretext tasks in the pre-training process of molecular representation learning. However, these methods have some disadvantages: (1) When the constructed motif dictionary is large, the computation of motif prediction is expensive. (2) Motifs as the self-supervised signal only, are difficult to be incorporated by hidden representations directly. We encode motifs contained in each molecule in the encoder part, which directly integrates the motif structure into the hidden representations. Moreover, the augmentation of motif-level nodes better captures the affiliation relationship between atoms and motifs.

**Graph encoding**. In addition to motif-level nodes, we also augment a graph-level node into the molecular graph, which

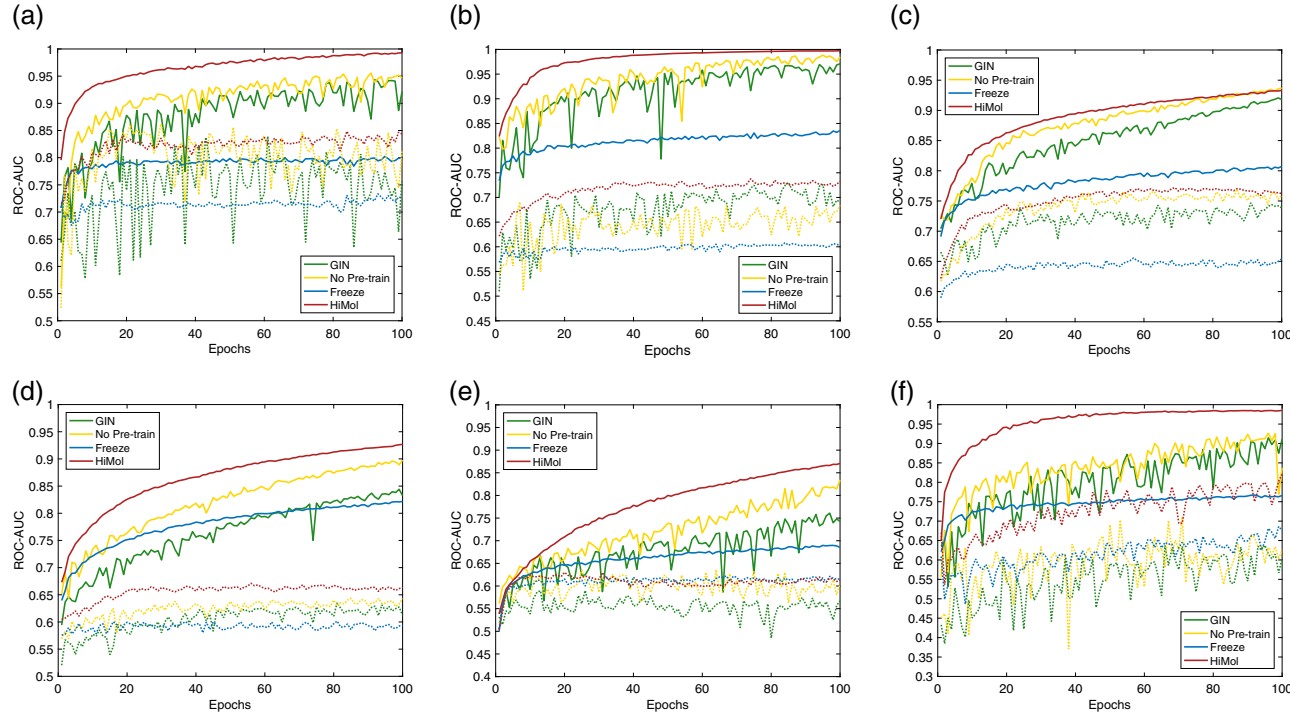

**Fig. 4 Training and testing classification ROC-AUC curves of different fine-tuning modes.** Solid lines are training curves; dashed lines are testing curves.

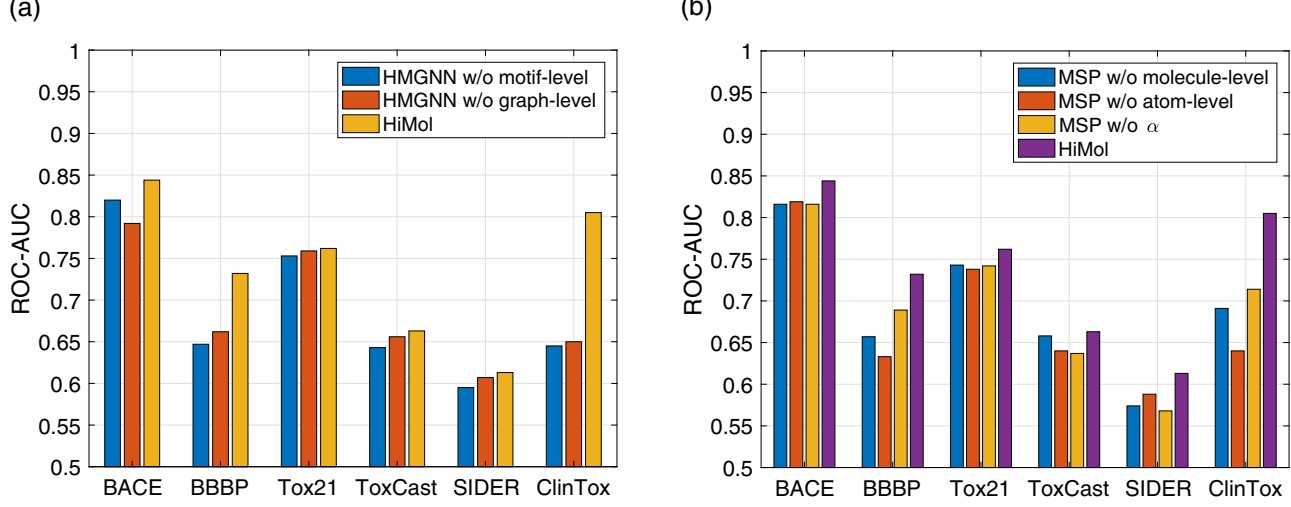

**Fig. 5 Ablation effect of HiMol pre-training on classification benchmarks. a** The performance of HMGNN with different ablations during pre-training. **b** The performance of MSP with different ablations during pre-training.

participates in the aggregation process to extract molecule representation. Most molecular learning methods use vanilla GNNs as the backbones and squeeze node representations into graph representation through the READOUT function, i.e., graph pooling methods like MAX, SUM, and MEAN. The process of GNNs is introduced in Supplementary Note 1. It is convenient to handle varying sizes of graph data, but the graph-level representation fails to be involved in learning. BERT model[11,44] proposes to attach a token to the sequence, whose representation is regarded as the sequence-level feature. Referring to BERT, Graphormer[45] adds a virtual node into the graph, which is linked and updated with all nodes, and the feature of the virtual node is the graph representation. Inspired by the success of the aforementioned works, we augment a graph-level node to extract the molecular graph representation via participating in aggregation

and updating with all atoms. Nevertheless, related methods only connect the graph-level node with normal nodes, they are not conducive to capturing subgraph structure. To integrate motif nodes, we connect graph-level nodes with motif nodes to achieve hierarchical aggregation. Compared with the READOUT, our HiMol implements the training and updating of graph nodes and realizes the interaction with multi-order nodes after incorporating molecular motif structure.

**Conclusion**

In this paper, we propose Hierarchical Molecular Graph Self-supervised Learning (HiMol) to learn molecular presentations. Specifically, we design Hierarchical Molecular Graph Neural Network (HMGNN) backbone to encode node-motif-graph hierarchical representations, which mines chemical semantic

information implied at motifs and realizes the interaction of multi-order features. Furthermore, we build multi-level molecular-dependent tasks as self-supervised signals, capturing more comprehensive information from multiple perspectives. Extensive molecular property prediction experiments demonstrate that HiMol shows great improvements over SOTA methods.

The superior performance of molecular representations reflects the potential of our HiMol framework, and there are interesting and promising future works on the basis of HiMol. (1) Combining molecular features with different dimensions, in addition to 2D graphs. (2) Simulating the generative process of molecules and extending our model to molecule generation and optimization tasks.

## Methods

**Hierarchical Molecular Graph Neural network.** We propose a Hierarchical Molecular Graph Neural network (HMGNN) to encode and represent molecular graphs, which mainly contains three parts: (1) motif construction; (2) augmented graph construction; (3) hierarchical representation encoder.

First, we decompose each input molecule graph into a group of motifs. BRICS algorithm is a traditional approach to fragment molecule graphs, which designs chemical rules to break bonds. Specifically, BRICS structurally decomposes molecules into multiple functional fragments according to whether bonds can be combined. However, BRICS splits molecules based on limited chemical rules, resulting in coarse granularity of decomposition. After BRICS decomposition, some large motifs still contain a few more general and functional substructures to be partitioned. Therefore, we add a decomposition rule on the basis of BRICS, i.e., decompose more than one ring substructures connected by common chemical bonds into minimum rings. Figure 6 illustrates the process of motif construction.

All obtained motifs are added as nodes $V_m$ to the molecular graph. Naturally, node-motif edges $E_m$ are added between each motif node and all the atom nodes it covers. Since the final output is the molecular graph representation, we construct a graph-level node $V_g$ into the molecular graph and link it to all motif nodes to form motif-graph edges $E_g$. The final augmented graph is represented as:

$$\widetilde{G} = (\widetilde{V}, \widetilde{E}), \widetilde{V} = [V, V_m, V_g], \widetilde{E} = [E, E_m, E_g] \tag{2}$$

On the constructed graph $\widetilde{G}$, we employ GNN to encode hierarchical molecular representations. Graph Isomorphism Network (GIN)[38] is the most widely used to encode molecular graph representation. Inspired by this, we apply GIN as the backbone model of our HiMol, the aggregation pattern of $l^{th}$ layer is given as follows:

$$\mathbf{h}_v^l = \mathrm{MLP}_a^l \left( \mathbf{h}_v^{l-1} + \sum_{u \in \mathcal{N}(v)} \left( \mathbf{h}_v^{l-1} + \mathrm{MLP}_b^l(\mathbf{X}_{uv}^b) \right) \right) \tag{3}$$

where $\mathrm{MLP}_a^l$ and $\mathrm{MLP}_b^l$ represent MLP for atom and bond feature transitions at $l$-layer, respectively, $\mathbf{X}_{uv}^b$ denotes the bond feature of $uv$, and $\mathbf{h}_v^0 = \mathbf{X}_v^a$ is the input atom feature of $v$. The detailed input features of atoms and bonds are shown in Supplementary Table 1. Different from other baselines, the initial molecular features of our HiMol contain degrees of atom nodes and whether bonds are in rings, which contain more molecular structural information. Our proposed HMGNN can encode molecular graphs to obtain multi-level representations simultaneously, including atom-level, motif-level, and molecule-level representations. The learned molecule-level representation can be directly used for downstream tasks, without the READOUT operation.

**Multi-level Self-supervised Pre-training.** After encoding the hierarchical representations through HMGNN, we design generative and predictive pretext tasks to perform Multi-level Self-supervised Pre-training (MSP). The types of self-supervised learning corresponding to different pretext tasks are described in Supplementary Note 2.

For atom-level representations, we design three generative tasks including bond links, atom types, and bond types, aiming at reconstructing graph structure and attributes. Specifically, three 2-layer MLPs with ReLU activation function project atom-level representations to predict three generative tasks, respectively. Cross entropy loss is employed to optimize the atom-level representations as follows:

$$L_{link} = -\sum_{v_i, v_j \in V} y_{ij} \log \hat{y}_{ij} + (1 - y_{ij}) \log(1 - \hat{y}_{ij}) \tag{4}$$

$$L_{atom\_type} = -\frac{1}{|V|} \sum_{v \in V} \sum_{k=1}^{K_{atom}} y_{v,k} \log \hat{y}_{v,k} \tag{5}$$

$$L_{bond\_type} = -\frac{1}{|E|} \sum_{e \in E} \sum_{k=1}^{K_{bond}} y_{e,k} \log \hat{y}_{e,k} \tag{6}$$

where $y_{ij}$ represents the link between $v_i$ and $v_j$, $y_{v,k} = 1$ represents the atom type of node $v$ is $k$, and $y_{e,k} = 1$ represents the bond type of edge $e$ is $k$. Their corresponding $\hat{y}$ are the predicted values. $K_{atom}$ and $K_{bond}$ are the number of atom types and bond types.

For graph-level representations, two predictive tasks on the overall properties of molecules are performed, i.e., the number of atoms and bonds. Similarly, two respective 2-layer MLPs with softplus activation function are applied to predict the graph-level representations. Based on SmoothL1 loss, the loss of predictive tasks are given in Equations (7) and (8).

$$L_{atom\_num} = \begin{cases} 0.5 \times (y_a - \hat{y}_a)^2, & if |(y_a - \hat{y}_a)| < 1 \\ |(y_a - \hat{y}_a)| - 0.5, & if |(y_a - \hat{y}_a)| \geq 1 \end{cases} \tag{7}$$

$$L_{bond\_num} = \begin{cases} 0.5 \times (y_b - \hat{y}_b)^2, & if |(y_b - \hat{y}_b)| < 1 \\ |(y_b - \hat{y}_b)| - 0.5, & if |(y_b - \hat{y}_b)| \geq 1 \end{cases} \tag{8}$$

where $y_a$ and $y_b$ are the number of atoms and bonds, respectively; $\hat{y}$ are predicted values. Supplementary Method describes the transformation process from the representation $\mathbf{H}$ to the predicted values $y$. Finally, we add a learnable vector weight $\alpha = [\alpha_1, \alpha_2, \alpha_3, \alpha_4, \alpha_5]$ normalized by softmax function to balance several losses. The overall objective is calculated as follows:

$$L = \alpha_1 L_{link} + \alpha_2 L_{atom\_type} + \alpha_3 L_{bond\_type} + \alpha_4 L_{atom\_num} + \alpha_5 L_{bond\_num} \tag{9}$$

**Datasets.** For the pre-training of HiMol, we utilize sampled 250K unlabeled molecules from the ZINC15[46] dataset. To evaluate the effectiveness of HiMol, we conduct molecular property prediction experiments on 12 datasets from MoleculeNet[47], including six classification task datasets and six regression task datasets. All downstream datasets are split into 80/10/10% for train/validation/test through scaffold-split, which is split in accordance with the molecular structures and provides more challenge for prediction tasks than random-split. The statistical data of all datasets are summarized in Supplementary Table 2.

**Baselines.** To evaluate the effectiveness of our proposed HiMol, different types of self-supervised learning SOTA are conducted as benchmarks. We compare diverse universal graph learning methods, including GraphSAGE[39], GPT_GNN[48], AttributeMask[49], ContextPred[49], InfoGraph[50], GraphLoG[26], GraphCL[25], and JOAO[51]. Furthermore, we compare several methods designed specifically for molecular graphs, i.e., MoCL[28], MolCLR[5], G_Motif[29], and MGSSL[30]. The detailed

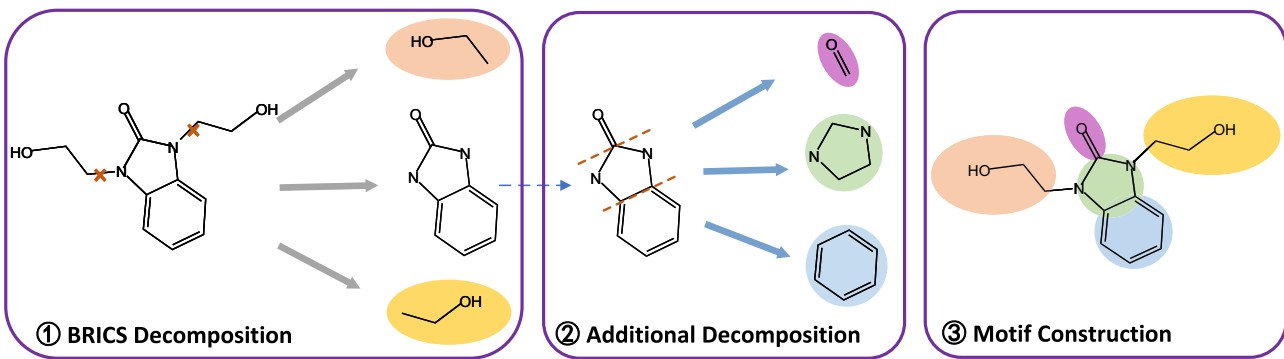

**Fig. 6 Overview of motif construction.** The construction process consists of three steps: (1) A given molecule graph is decomposed according to BRICS. (2) The molecule is further decomposed through our additional rule. (3) Decomposed motifs are constructed as motif-level nodes.

① BRICS Decomposition   ② Additional Decomposition   ③ Motif Construction

description of baselines is demonstrated in Supplementary Note 3, and the SSL type is summarized in Supplementary Table 3.

**Experimental configuration**. All experiments are conducted on the Linux server with Nvidia GeForce GTX 1080 Ti GPU and Intel(R) Xeon(R) CPU E5-2620 v4 @ 2.10GHz. In the pre-training phase, the HiMol run 100 epochs using the Adam optimizer with a learning rate of 0.001. We train the LARGE and SMALL versions, with 3-layer and 5-layer GIN as the HMGNN backbone. The dropout rate for the GIN backbone is set to 0.5. The batch size is set as 32 and the number of workers for dataloader is 8. The hidden representations of all levels are 512-dimension.

In the fine-tuning phrase, we implement a 2-layer MLP with ELU activation function as the classifier. The weights of both pre-trained HMGNN and the classifier are optimized by the Adam optimizer with respective learning rates ranging from 1e-4 to 1e-3. For each downstream task, we fine-tune for 100 epochs and report the average test results of five runs. The batch size is 32 and the number of workers is 4. The embedding dimension is 512, the same as that in the pre-training phase. The dropout rate is adjusted in the range [0.5, 0.7]. The parameter settings for each dataset during the fine-tuning process are reported in Supplementary Table 4.

For the purpose of fairness, we pre-train all the baselines on the same dataset ZINC15 according to their official codes, except MoCL[28]. MoCL does not implement transfer learning, and it is computationally expensive to calculate the similarity matrix of ZINC15 dataset during pre-training. Therefore, we adopt the same manner as the official work: pre-training on each downstream dataset and then fine-tuning on the same dataset for molecular property prediction tasks. All baselines maintain the same split proportion and manner on the downstream datasets.

## Data availability
All related data in this paper are public. The ZINC dataset for pre-training can be downloaded from https://github.com/zaixizhang/MGSSL/tree/main/motif_based_pretrain/data/zinc as described in MGSSL[30]. All downstream datasets for fine-tuning can be downloaded from MoleculeNet website https://github.com/deepchem/deepchem/tree/master/deepchem/molnet/load_function.

## Code availability
The implementation of HiMol is publicly available at https://github.com/ZangXuan/HiMol.

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

## Acknowledgements

This study is partially supported by the National Natural Science Foundations of China (62276082 and 61876052), Major Key Project of PCL (PCL2021A06), Strategic Emerging Industry Development Special Fund of Shenzhen (20200821174109001) and Pilot Project in 5G + Health Application of Ministry of Industry and Information Technology & National Health Commission (5G + Luohu Hospital Group: an Attempt to New Health Management Styles of Residents).

## Author contributions

Xu.Z. proposed the research, conducted experiments, analyzed the data, and wrote the manuscript. Xi.Z. improved the manuscripts. B.T. supervised the overall project.

## Competing interests

The authors declare no competing interests.
