## [Peer Review File · Communications Chemistry]

Reviewers' comments:

Reviewer #1 (Remarks to the Author):

In this work, the authors proposed HiMol, which contains a hierarchical GNN (HMGNN) and multi-level self-supervised pre-training to learn representations from molecular graphs. Experiments shown that HiMol pre-trained on 250K data from ZINC15 outperformed self-supervised learning baselines on most MoleculeNet benchmarks. Additionally, ablation study was conducted to unveil the effectiveness of both the hierarchical architecture and self-supervised pre-training. The paper can be an interesting contribution to the area of molecular graph representation learning. However, I have the following concerns.

This work introduces hierarchical GNN and multi-level pre-training to explicitly learn information of motifs in molecules. However, some previous works have also investigated molecular self-supervised learning in multi-levels, e.g., Zhang et al. (<https://arxiv.org/abs/2110.00987>), Wang et al. (<https://pubs.acs.org/doi/full/10.1021/acs.jcim.2c00495>), and Zhang & Hu et al. (<https://arxiv.org/abs/2012.12533>). I suggest the authors discuss the works in Introduction and emphasize the difference between HiMol and previous works in multi-level learning.

In HMGNN, motif nodes are not directly connected. Is there a reason why motif connections are not included in the hierarchical molecular graph?

In atom number and bond number predictions, the problem is considered as regression and trained via smooth L1 loss. Why not modeling the tasks as classifications? Did the authors by any chance try the classification settings?

Figure 2 shows the fine-tuned HiMol representations in 4 datasets. It may better illustrate the effectiveness of HiMol by visualizing representations from unpretrained models side-by-side for comparison.

In Equation 9, the authors mention learnable coefficients α for different loss terms. How is α learned during training?

Reviewer #2 (Remarks to the Author):

The paper presents a novel hierarchical graph neural network to encode molecular motif structures and extract node-motif-graph (atom-motif-molecule) representations, and a corresponding multi-level self-supervised pre-training framework with increased transferability. The idea is interesting, the figures are of good quality, and overall, the paper is well written.

My main confusion is the Hierarchical Molecular Graph Neural network (HMGNN).

1. In the methods section (page 13), the authors mention that nodes and edges from three levels (atomic, motif and graph) are merged into one big graph, and a Graph Isomorphism Network (GIN) is then used to predict features such as node-level representations. The illustration in Figure 1 shows a different structure. It seems that there are multiple GNNs to analyze the graph at three levels.

2. Also, if you have merged all three levels, how do you differentiate the features of the different levels?

3. I see edges between nodes and motifs, are there edges between motifs? If not, is there any reason to ignore motif interactions?

4. Can you elaborate on the equation jump from 3 (page 13) to 4-8 (page 14). If you do not have a READOUT operation, how do you connect h in equation 3 to any of the elements in 4-8 for the loss function?

5. Page 13, line 369 and Table 4. The node feature contains only atomic index and atomic degree. Why choose these two simple features? Why not include other atomic features, such as the Weave

features using Deepchem.

Other questions I have are

6. Page 4, line 147. How did you handle different number of atoms and motifs in a molecule? Did you use batched graph or masking?

7. Figure 1. It is desirable to standardize the formatting. For example, some words have spaces filled with underscores (e.g., node_level), while in the title they are dashes (node-level).

8. Can you comment on the results in Tables 1 and 2 against MoleculeNet benchmarks? I see that the best performing results are not as good as the MoleculeNet benchmark. Is it because the features are from a self-supervised learning method or is it because of the choice of the test set?

9. A general question about motif feature selection. I see that motifs often include functional groups that are critical to chemical properties. Why not include functional group features? On page 11, you mention that "When the constructed motif dictionary is large, the computation of motif prediction is expensive". Can you comment on the size of the motif dictionary and how this would make the computation expensive?

Reviewer #1:

In this work, the authors proposed HiMol, which contains a hierarchical GNN (HMGNN) and multi-level self-supervised pre-training to learn representations from molecular graphs. Experiments shown that HiMol pre-trained on 250K data from ZINC15 outperformed self-supervised learning baselines on most MoleculeNet benchmarks. Additionally, ablation study was conducted to unveil the effectiveness of both the hierarchical architecture and self-supervised pre-training. The paper can be an interesting contribution to the area of molecular graph representation learning. However, I have the following concerns.

1. This work introduces hierarchical GNN and multi-level pre-training to explicitly learn information of motifs in molecules. However, some previous works have also investigated molecular self-supervised learning in multi-levels, e.g., Zhang et al. (<https://arxiv.org/abs/2110.00987>), Wang et al. (<https://pubs.acs.org/doi/full/10.1021/acs.jcim.2c00495>), and Zhang & Hu et al. (<https://arxiv.org/abs/2012.12533>). I suggest the authors discuss the works in Introduction and emphasize the difference between HiMol and previous works in multi-level learning.

Many thanks for providing the related studies. We have carefully reviewed and considered the papers provided by the reviewer, the differences between these methods and our HiMol are summarized as follows.

(1) For Zhang, Z., Liu, Q., Wang, H., Lu, C., Lee, C.-K.: Motif-based graph self-supervised learning for molecular property prediction. *Advances in Neural Information Processing Systems* 34, 15870–15882 (2021).

(<https://arxiv.org/abs/2110.00987>)

The authors proposed a self-supervised learning framework based on motif to learn molecular representation. However, the encoder only uses the local topology structure between atoms to get the hidden atom-level representations, which are pooled to obtain the molecular representation through READOUT. Our HiMol incorporates the motif structure and augments the graph-level node in the encoder, which are conducive to learning hierarchical molecular representations. On the other hand, Zhang et al. proposed motif prediction as pretext tasks, where the large motif dictionary makes the time computation expensive. We design multi-level tasks to pre-train, which improve the transferable ability of model.

(2) For Wang, Y., Magar, R., Liang, C., Barati Farimani, A.: Improving molecular contrastive learning via faulty negative mitigation and decomposed fragment contrast. *Journal of Chemical Information and Modeling* (2022).

(<https://pubs.acs.org/doi/full/10.1021/acs.jcim.2c00495>)

The paper proposed a contrastive learning framework for molecular learning, in which not only molecular pairs but also motif pairs are sampled for contrastive pre-training. Differently, our pre-training framework HiMol is a hybrid of self-supervised strategies including generative

and predictive learning. The mixed self-supervised learning of multiple tasks increases the generalization performance of model pre-training. In addition, the motif structure in this work is only used in decoder as in the paper (1).

(3) For Zhang, S., Hu, Z., Subramonian, A., Sun, Y.: Motif-driven contrastive learning of graph representations. arXiv preprint arXiv:2012.12533 (2020).

(<https://arxiv.org/abs/2012.12533>)

The paper modeled a clustering problem to learn molecular motifs, and the GNN-encoded atom representations are grouped into subgraphs for contrastive learning. The motif is learned through clustering without introducing chemical prior knowledge. The effectiveness of motif generation is more likely to be affected by training data. As for our HiMol, motif is constructed based on the rule of BRICS, so that the generated motifs are in line with chemical functional groups and more capable of reflecting chemical properties.

In summary, though the aforementioned three works are motif-based molecular learning methods, our framework HiMol has some innovations. (1) We mine motif structure during encoding molecular representations, making motifs more directly involved in molecular learning in both pre-training and fine-tuning. (2) We design multi-level hybrid self-supervised learning to pre-train, which improve the transferable performance. Related content has been summarized and cited to enrich the content of Introduction (Page 2).

2. In HMGNN, motif nodes are not directly connected. Is there a reason why motif connections are not included in the hierarchical molecular graph?

Motif nodes are added to mine the topology structure of molecular functional groups composed of atoms and better conduct the interaction between atom-level and graph-level representations as intermedium. Theoretically, the reasons why motif connections are not considered are as follows. Firstly, the motif nodes are virtual, then adding edges between the motif virtual nodes may introduce noise. Secondly, edges between motifs are derived from the original bonds that already exist at the atom-level nodes, so adding repetitive edges is meaningless for augmented graph construction.

We conducted experiments to add connections between motif pairs with shared atoms or bonds. The type of motif-motif edges distinguishes the four types of molecular bonds as well as the node-motif and motif-graph edges. The results are shown in the table, where HiMol_Motif is the version with motif connections, and the number of encoder layers of HiMol and HiMol_Motif is 5. Overall, HiMol_Motif and HiMol have similar performance, which implies that motif connections have no significant effect on the molecular learning.

Dataset	BACE	BBBP	Tox21	ToxCast	SIDER	CLINTOX
HiMol	84.3	73.2	76.2	66.3	61.3	80.8
HiMol_Motif	84.5	71.4	76.6	64.8	62.3	79.7

Dataset	ESOL	FreeSolv	Lipophilicity	QM7	QM8	QM9
---------	------	----------	---------------	-----	-----	-----

Metrics	RMSE	RMSE	RMSE	MAE	MAE	MAE
HiMol	0.833	2.283	0.708	91.501	0.0199	3.243
HiMol_Motif	0.813	2.417	0.691	100.44	0.0191	3.447

3. In atom number and bond number predictions, the problem is considered as regression and trained via smooth L1 loss. Why not modeling the tasks as classifications? Did the authors by any chance try the classification settings?

We supplemented the graph-level pretext tasks of predicting atom number and bond number as classification tasks. Firstly, the number of atoms and bonds of all molecules in pre-training dataset ZINC are counted, and then the categories of their numbers are set as the dimension of the last layer of MLPs for prediction. The MLP is two layers with RELU activation function. The loss function is the Cross-Entropy loss.

The results are shown in the table below, where HiMol_Cla is the version with classification settings and the backbone is 5-layer GIN. The classification setting HiMol_Cla has worse performance than our regression setting HiMol in most cases. In addition, the categories of atom number and the bond number are different in other datasets. It is necessary to calculate and alter the dimension of the neural networks when changing pre-training datasets, leading to low generalization.

Dataset	BACE	BBBP	Tox21	ToxCast	SIDER	CLINTOX
HiMol	84.3	73.2	76.2	66.3	61.3	80.8
HiMol_Cla	77.7	69.0	75.3	66.4	59.9	72.8

Dataset	ESOL	FreeSolv	Lipophilicity	QM7	QM8	QM9
Metrics	RMSE	RMSE	RMSE	MAE	MAE	MAE
HiMol	0.833	2.283	0.708	91.501	0.0199	3.243
HiMol_Cla	0.890	2.305	0.752	309.646	0.0208	3.458

4. Figure 2 shows the fine-tuned HiMol representations in 4 datasets. It may better illustrate the effectiveness of HiMol by visualizing representations from unpretrained models side-by-side for comparison.

According to the suggestions of reviewers, we supplement the visualization of molecular representations without pre-training. It can be observed from the figure below, the representation of atoms without pre-training is more chaotic. The visualization results after pre-training in the manuscript (Fig.2) show that the molecular representations learn more chemical properties. This illustrates the effectiveness of our pre-training framework. We updated the manuscript by supplementing the above results and analysis of experiments into Appendix C.2.

5. In Equation 9, the authors mention learnable coefficients α for different loss terms. How is α learned during training?

We define a variable $\alpha = [\alpha_1, \alpha_2, \alpha_3, \alpha_4, \alpha_5] \in \mathbb{R}^5$ with gradient calculation (`requires_grad=True`). After obtaining the loss of each prediction task ($L_{link}, L_{atom_type}, L_{bond_type}, L_{atom_num}, L_{bond_num}$), we use α to add all the loss to obtain the overall loss L through equation 9:

$$L = \alpha_1 L_{link} + \alpha_2 L_{atom_type} + \alpha_3 L_{bond_type} + \alpha_4 L_{atom_num} + \alpha_5 L_{bond_num}$$

When the overall loss L is backpropagated, the gradient of α is calculated and α is updated. Therefore, α is learned during the model training.

Reviewer #2:

The paper presents a novel hierarchical graph neural network to encode molecular motif structures and extract node-motif-graph (atom-motif-molecule) representations, and a corresponding multi-level self-supervised pre-training framework with increased transferability. The idea is interesting, the figures are of good quality, and overall, the paper is well written.

My main confusion is the Hierarchical Molecular Graph Neural network (HMGNN).

1. In the methods section (page 13), the authors mention that nodes and edges from three levels (atomic, motif and graph) are merged into one big graph, and a Graph Isomorphism Network (GIN) is then used to predict features such as node-level representations. The illustration in Figure 1 shows a different structure. It seems that there are multiple GNNs to analyze the graph at three levels.

The augmented molecular graph merging three-level nodes (atom, motif and graph) is input into an L-layer GNN to encode the hidden representations of the three-level nodes simultaneously.

To further clarify our method, we revised the framework of HMGNN. Mainly, we figure the augmented molecular graph, in which the white nodes represent atoms, the colored ones denote decomposed motifs, and the gray node is the molecule. In addition, cross-level edges (node-motif and motif-graph) are added. The augmented graph is input into the L-layer GNN to extract the node-level, motif-level, and graph-level hidden representations jointly.

2. Also, if you have merged all three levels, how do you differentiate the features of the different levels?

We demonstrate how to differentiate node features of different levels in terms of input features, the encoding and decoding process for hidden representations.

For input features, we use atomic index and atomic degree to initialize node features. Graph-level and motif-level are indexed 119 and 120 after all the atomic indices. Their degrees are set to 0, which is distinct from the fact that the atomic degree is greater than zero.

For the encoding process for hidden representations, nodes aggregate the neighborhood information to obtain the hidden representations. Different levels of nodes aggregate different sources of information. For example, atom nodes aggregate information from neighborhood atoms and the motif node, motif nodes aggregate from the graph node and its contained atom nodes, and the graph node aggregates all motif node features.

For the decoding process for hidden representations, we use the *num_part* built when we process the input molecular graph to divide nodes of different levels. The *num_part* is a list of the number of atom-level, motif-level and graph-level nodes in each molecule, i.e., $[|V|, |V_m|, 1]$. According to the *num_part*, we split all node hidden representations $H \in \mathbb{R}^{(|V|+|V_m|+1) \times d}$ to obtain atom-level representations $H_a \in \mathbb{R}^{|V| \times d}$ and graph-level representation $H_g \in \mathbb{R}^d$ for two-level prediction tasks of the decoder.

3. I see edges between nodes and motifs, are there edges between motifs? If not, is there any reason to ignore motif interactions?

Motif nodes are added to mine the topology structure of molecular functional groups composed of atoms and better conduct the interaction between atom-level and graph-level representations as intermedium. Theoretically, the reasons why motif connections are not considered are as follows. Firstly, the motif nodes are virtual, then adding edges between the motif virtual nodes may introduce noise. Secondly, edges between motifs are derived from the original bonds that already exist at the atom-level nodes, so adding repetitive edges is meaningless for augmented graph construction.

We conducted experiments to add connections between motif pairs with shared atoms or bonds. The type of motif-motif edges distinguishes the four types of molecular bonds as well as the node-motif and motif-graph edges. The results are shown in the table, where HiMol_Motif is the version with motif connections, and the number of encoder layers of HiMol and HiMol_Motif is 5. Overall, HiMol_Motif and HiMol have similar performance, which implies that motif connections have no significant effect on the molecular learning.

Dataset	BACE	BBBP	Tox21	ToxCast	SIDER	CLINTOX
HiMol	84.3	73.2	76.2	66.3	61.3	80.8
HiMol_Motif	84.5	71.4	76.6	64.8	62.3	79.7

Dataset	ESOL	FreeSolv	Lipophilicity	QM7	QM8	QM9
Metrics	RMSE	RMSE	RMSE	MAE	MAE	MAE
HiMol	0.833	2.283	0.708	91.501	0.0199	3.243
HiMol_Motif	0.813	2.417	0.691	100.44	0.0191	3.447

4. Can you elaborate on the equation jump from 3 (page 13) to 4-8 (page 14). If you do not have a READOUT operation, how do you connect h in equation 3 to any of the elements in 4-8 for the loss function?

In equation 3, h represents the update rule of each node of all levels. We utilize atom representations $H_a \in \mathbb{R}^{|V| \times d}$ and graph representation $H_g \in \mathbb{R}^d$ to conduct atom-level and molecule-level prediction tasks in the decoder respectively.

In equation 4, $\hat{y}_{ij} = \phi_{link}(concat[h_i, h_j])$, where $i, j \in V$, and ϕ_{link} denotes $\{Linear(2d, d) \rightarrow Relu \rightarrow Linear(d, 1)\}$.

In equation 5, $\hat{y}_{v,k} = \phi_{atom_type}(h_v)$, where $v \in V$, and ϕ_{atom_type} denotes $\{Linear(d, d) \rightarrow Relu \rightarrow Linear(d, N_{atom_type})\}$, $N_{atom_type} = 118$.

In equation 6, $\hat{y}_{e,k} = \phi_{bond_type}(concat[h_i, h_j])$, where $e = ij$, $i, j \in V$, and ϕ_{bond_type} denotes $\{Linear(2d, d) \rightarrow Relu \rightarrow Linear(d, N_{bond_type})\}$, $N_{bond_type} = 4$.

In equation 7, $\hat{y}_a = \phi_{atom_num}(H_g)$, where ϕ_{atom_num} denotes $\{Linear(d, d/4) \rightarrow Softplus \rightarrow Linear(d/4, 1)\}$.

In equation 8, $\hat{y}_b = \phi_{bond_num}(H_g)$, where ϕ_{bond_num} denotes $\{Linear(d, d/4) \rightarrow Softplus \rightarrow Linear(d/4, 1)\}$.

We supplemented these formations of five MLPs in Appendix B.

5. Page 13, line 369 and Table 4. The node feature contains only atomic index and atomic degree. Why choose these two simple features? Why not include other atomic features, such as the Weave features using Deepchem.

We only chose two atomic attributes for initial features, where atomic type is the most basic atomic characteristic, and atomic degree is the most basic topological feature. Thanks to the reviewer's suggestion, we trained with HiMol with the Weave features as the initial feature. From the results in table, our model HiMol with two features has superior performance on most datasets. Interestingly, the classification result of HiMol_Weave has significant improvements on the ClinTox dataset, even exceeding the best comparison algorithms in the manuscript. It can be explained that the extra features of Weave features are important for the molecules of ClinTox.

Dataset	BACE	BBBP	Tox21	ToxCast	SIDER	CLINTOX
HiMol	84.3	73.2	76.2	66.3	61.3	80.8
HiMol_Weave	83.5	69.7	76.2	65.4	60.9	92.1

Dataset	ESOL	FreeSolv	Lipophilicity	QM7	QM8	QM9
Metrics	RMSE	RMSE	RMSE	MAE	MAE	MAE
HiMol	0.833	2.283	0.708	91.501	0.0199	3.243
HiMol_weave	0.890	2.351	0.644	91.084	0.0194	3.312

Other questions I have are

6. Page 4, line 147. How did you handle different number of atoms and motifs in a molecule? Did you use batched graph or masking?

We import Data and Batch from torch_geometric.data to construct batched graph without masking. Firstly, each molecule is constructed as an object of the Data class, including the atom attributes, the edge index, the edge attribute, and num_part. Then all Data objects of molecules within a batch are placed in a list and then constructed into a Batch object. The batch object constructs a big graph (disconnected) containing all molecular graphs within the batch. Specifically, the Batch class uniformly reindexes all nodes of each molecular graph, that is, the index of the node is updated sequentially in the order of the molecular graphs. The reindex is formalized as $id_p^{new} = \sum_{k < p} |V|_k + id_p^{old}$, where id_p denotes the index of nodes of the graph p in the batch, and $|V|_k$ is the number of nodes of the graph k . The Batch class can process molecules with different number of nodes for a batched graph without masking. The torch_geometric.data.Data¹ and torch_geometric.data.Batch² can refer to the official code.

1.https://pytorch-geometric.readthedocs.io/en/latest/_modules/torch_geometric/data/data.html

2.https://pytorch-geometric.readthedocs.io/en/latest/_modules/torch_geometric/data/batch.html

7. Figure 1. It is desirable to standardize the formatting. For example, some words have spaces filled with underscores (e.g., node_level), while in the title they are dashes (node-level).

Special thanks to the reviewer for finding the errors of the formatting. We have carefully checked inconsistent formatting issues in this manuscript. In detail, we updated the Node_level, Motif_level and Graph_level to Node-level, Motif-level and Graph-level in Figure 1 (Page 3).

8. Can you comment on the results in Tables 1 and 2 against MoleculeNet benchmarks? I see that the best performing results are not as good as the MoleculeNet benchmark. Is it because the features are from a self-supervised learning method or is it because of the choice of the test set?

The way of splitting test set has a great influence on the performance. Most of the downstream datasets in MoleculeNet benchmark^[1] utilize the random splitting. However, our experiments split the data via scaffold splitting, in line with the recent state-of-the-arts. The scaffold splitting increases the challenge of the downstream tasks.

We supplemented the experimental results of random splitting in the table below, in which we set three different random seeds. ↑ implicates the higher the value, the better the performance, while ↓ means the lower the value, the better the performance. It can be observed from the results, (1) Our results are better than MoleculeNet benchmarks overall; (2) The result of random splitting will fluctuate greatly with the change of seed, especially on small-scale data sets (like ClinTox, SIDER, ESOL and FreeSolv). The detailed settings of random

seeds is not shown in the MoleculeNet benchmark^[1]. Therefore, the random splitting is unstable, so that it is difficult to ensure the fairness of comparison.

As for BACE and BBBP, MoleculeNet benchmark also used the scaffold splitting. Our performance on BACE is slightly worse than the best performance of conventional methods (RF:86.7) and better than the best graph-based methods (Weave: 80.6). It was demonstrated in the MoleculeNet benchmark^[1] that the graph-based methods have sparse training data on the small-scale datasets, resulting in the learned representations that are less robust for specific tasks than the conventional methods. Our results on BBBP are better than all methods of MoleculeNet benchmarks.

	HiMol Seed=0	HiMol Seed=42	HiMol Seed=100	Best performances - conventional methods of MoleculeNet benchmark	Best performances - graph-based methods of MoleculeNet benchmark
ClinTox(↑)	83.9	75.8	83.3	82.7	83.2
SIDER(↑)	64.6	61.9	68.5	68.4	63.8
Tox21(↑)	85.1	84.5	85.8	82.2	82.9
ToxCast(↑)	75.1	75.1	74.0	70.2	74.2
ESOL(↓)	0.774	0.607	0.595	0.99	0.58
FreeSolv(↓)	2.261	1.472	0.981	1.74	1.15
Lipophilicity(↓)	0.576	0.658	0.578	0.799	0.655
Qm8(↓)	0.0125	0.0127	0.0122	0.0150	0.0143
Qm9(↓)	2.790	2.293	2.756	4.35	2.35

[1] Wu, Z., Ramsundar, B., Feinberg, E.N., Gomes, J., Geniesse, C., Pappu, A.S., Leswing, K., Pande, V.: Moleculenet: a benchmark for molecular machine learning. *Chemical science* 9(2), 513–530 (2018)

9. A general question about motif feature selection. I see that motifs often include functional groups that are critical to chemical properties. Why not include functional group features? On page 11, you mention that “When the constructed motif dictionary is large, the computation of motif prediction is expensive”. Can you comment on the size of the motif dictionary and how this would make the computation expensive?

To demonstrate the high computation of motif prediction, we compare MGSSL^[2] with our method HiMol that have the same data processing approach and the same GNN backbone. The time complexity of motif prediction is $O(K_{motif}|V_m|)$, and that of our MSP decoder prediction is $O(|V_a|K_{atom} + |E|K_{bond} + |V_a|^2 + 1 + 1)$, where K_{atom} and K_{bond} , are respectively the number of atom and bond types, $|V_a|$, $|V_m|$ and $|E|$ are respectively the average number atoms, motifs and bonds in a molecule, and K_{motif} is the

size of motif vocabulary. We calculate these values on the dataset ZINC for pre-training, which are shown in the table below. Since the value of K_{motif} is much larger than other values, the time complexity of motif prediction will be expensive. According to different methods of motif construction, if the larger the size of motif dictionary is, the higher the complexity of the algorithm.

K_{atom}	K_{bond}	K_{motif}	$ V_a $	$ E $	$ V_m $
118	4	$\sim 10^4$	~ 23	~ 25	~ 6

We also compare the running time of our method HiMol with that of MGSSL^[2], where we set the same batch_size and run them on the same Linux server with Nvidia GeForce GTX 1080 Ti GPU and Intel(R) Xeon(R) CPU E5-2620 v4 @ 2.10GHz. The table below shows the running time of each epoch.

HiMol (ours)	MGSSL
5,796s	22,280s

[2] Zhang, Z., Liu, Q., Wang, H., Lu, C., Lee, C.-K.: Motif-based graph self-supervised learning for molecular property prediction. Advances in Neural Information Processing Systems 34, 15870–15882 (2021)

REVIEWERS' COMMENTS:

Reviewer #2 (Remarks to the Author):

After reading through the revised manuscript, I am pleased to say that the authors have successfully addressed the issues and concerns raised in the previous review. The revised manuscript is well-written and clearly presents the research and its findings. I recommend publishing the revised manuscript.

#####

I am taking over for Reviewer #1 in reviewing the revised manuscript. Overall, the authors have successfully addressed all of Reviewer #1's comments and questions. The revised manuscript is well-written, and the figures and content are of high quality. The details are as follows:

For the first comment of Reviewer #1, the authors have effectively addressed, incorporating additional relevant literature into the introduction and demonstrating the innovative nature of the HiMol framework.

Regarding Reviewer #1's second comment, the authors have provided a rationale for not including motif connections and conducted additional experiments comparing results with and without such connections, which showed no significant difference. I find the response to be satisfactory.

In response to Reviewer #1's third comment, the authors have explained why they did not model the tasks as classifications and conducted additional experiments demonstrating that doing so resulted in worse performance. I agree with the authors' explanation and also think regression task is more appropriate.

Reviewer #1's fourth comment has been addressed through the inclusion of a new comparison, which demonstrates the need for the proposed pre-training framework.

For the fifth comment of Reviewer #1, the authors have provided a good explanation of how alpha is learned during training.